# Assessing the effect of closed-loop insulin delivery from onset of type 1 diabetes in youth on residual beta-cell function compared to standard insulin therapy (CLOuD study): a randomised parallel study protocol

Charlotte Boughton  ,[1] Janet M Allen,[1] Martin Tauschmann,[1] Sara Hartnell,[1] Malgorzata E Wilinska,[1] Gianluca Musolino,[1] Carlo L Acerini,[2] Professor David Dunger  ,[2] Fiona Campbell,[3] Atrayee Ghatak,[4] Tabitha Randell,[5] Rachel Besser,[6,7] Nicola Trevelyan,[8] Daniela Elleri,[9] Elizabeth Northam,[10] Korey Hood,[11] Eleanor Scott,[12] Julia Lawton,[13] Stephane Roze,[14] Judy Sibayan,[15] Craig Kollman,[15] Nate Cohen,[15] John Todd,[16] Roman Hovorka,[1] on behalf of CLOuD Consortium

For numbered affiliations see end of article.

**Correspondence to**
Dr Roman Hovorka;
rh347@cam.ac.uk

## ABSTRACT

**Introduction** Management of newly diagnosed type 1 diabetes (T1D) in children and adolescents is challenging for patients, families and healthcare professionals. The objective of this study is to determine whether continued intensive metabolic control using hybrid closed-loop (CL) insulin delivery following diagnosis of T1D can preserve C-peptide secretion, a marker of residual beta-cell function, compared with standard multiple daily injections (MDI) therapy.

**Methods and analysis** The study adopts an open-label, multicentre, randomised, parallel design, and aims to randomise 96 participants aged 10–16.9 years, recruited within 21 days of diagnosis with T1D. Following a baseline mixed meal tolerance test (MMTT), participants will be randomised to receive 24 months treatment with conventional MDI therapy or with CL insulin delivery. A further 24-month optional extension phase will be offered to all participants to continue with the allocated treatment. The primary outcome is the between group difference in area under the stimulated C-peptide curve (AUC) of the MMTT at 12 months post diagnosis. Analyses will be conducted on an intention-to-treat basis. Key secondary outcomes are between group differences in time spent in target glucose range (3.9–10 mmol/L), glycated haemoglobin (HbA1c) and time spent in hypoglycaemia (<3.9 mmol/L) at 12 months. Secondary efficacy outcomes include between group differences in stimulated C-peptide AUC at 24 months, time spent in target glucose range, glucose variability, hypoglycaemia and hyperglycaemia as recorded by periodically applied masked continuous glucose monitoring devices, total, basal and bolus insulin dose, and change in body weight. Cognitive, emotional and behavioural characteristics of participants and parents will be evaluated, and a cost–utility analysis performed to support adoption of CL as a standard treatment modality following diagnosis of T1D.

**Ethics and dissemination** Ethics approval has been obtained from Cambridge East Research Ethics Committee. The results will be disseminated by peer-reviewed publications and conference presentations.

**Trial registration number** NCT02871089; Pre-results.

## Strengths and limitations of this study

► The study adopts an open-label, multicentre, randomised, parallel design.
► The study includes youth newly diagnosed with type 1 diabetes.
► The study includes a 4-year follow-up period with equal numbers of study visits between both groups.
► The comparator group can use pump therapy and/ or flash or continuous glucose monitoring devices or closed-loop systems if clinically appropriate.
► The study includes psychosocial assessments and health economic analysis to support adoption of closed-loop systems in this population.

## INTRODUCTION

Type 1 diabetes (T1D) is characterised by insulin deficiency due to immune-mediated pancreatic beta-cell destruction.[1] Incidence of T1D is increasing, particularly among children.[2] Management of newly diagnosed T1D in children and adolescents is challenging for patients, families and healthcare professionals. Furthermore, glycaemic control commonly deteriorates in adolescence due

to physiological and psychological challenges relating to puberty. Most adolescents are unable to meet International Society for Paediatric and Adolescent Diabetes (ISPAD) target glycated haemoglobin (HbA1c) <7.0% (53 mmol/mol).[3] Fear of hypoglycaemia is common and affects quality of life and psychological well-being of children and families, leading to suboptimal glycaemic control.[4]

At clinical diagnosis of diabetes, most people retain some pancreatic islet cells that continue to secrete endogenous insulin for several years as reflected by C-peptide levels. In the Diabetes Control and Complications Trial, assignment to intensive therapy reduced the risk of loss of stimulated C-peptide by 57% over the mean 6.5-year study duration, showing that metabolic control was associated with preservation of islet cell function.[5] Those with C-peptide levels ≥0.20 pmol/mL initially or sustained over a year had fewer microvascular complications and fewer severe hypoglycaemic events.[6] A linear relationship has been shown between frequency of retinopathy progression and plasma C-peptide as low as 0.03 pmol/mL.[7]

Emergence of new technologies including continuous glucose monitoring,[8] sensor-augmented pump therapy[9] and threshold pump suspend[10 11] provides opportunities to improve outcomes. The most promising approach is a closed-loop (CL) system which combines real-time glucose monitoring with computer-based algorithm-directed insulin delivery to achieve glucose-responsive subcutaneous insulin delivery mimicking beta-cell function.[12] The first commercially available CL system, the MiniMed 670G pump (Medtronic, Northridge, California), was launched in the USA in 2017 and in Europe in 2018.

The CL approach has been evaluated in children and adolescents in controlled laboratory studies[13–15] and home settings.[16–20] The results demonstrate improved glucose control and reduced risk of hypoglycaemia events. Psychosocial assessments support acceptability among children/adolescents and carers.[21]

We hypothesise that the CL approach using the Cambridge CL algorithm can preserve residual beta-cell function. The present study will assess the impact of CL insulin delivery after diagnosis on C-peptide secretion. We will also assess feasibility and acceptance of this therapy to support adoption as a standard treatment modality following diagnosis of T1D.

## METHODS AND ANALYSIS
### Overview
The study adopts an open-label, multicentre, randomised, single-period, parallel design to assess the effect of CL insulin delivery using the Cambridge CL algorithm from onset of T1D in youth on residual beta-cell function compared with standard insulin therapy (figure 1). Participants will include youths aged 10–16.9 years diagnosed with T1D within the previous 21 days. The study aims to randomise 96 participants. Recruited participants will be randomly assigned to

24 months of study intervention. There will be an optional extension phase of a further 24 months of study intervention online supplementary appendix.

The University of Cambridge (UK) will be the coordinating centre. Clinical sites include:
1. Addenbrooke's Hospital, Cambridge.
2. Leeds Children's Hospital, Leeds.
3. Alder Hey Children's Hospital, Liverpool.
4. Nottingham Children's Hospital, Nottingham.
5. Oxford Children's Hospital, Oxford.
6. Southampton Children's Hospital, Southampton.
7. Royal Hospital for Sick Children, Edinburgh.

Participants may also be recruited from Patient Identification Centres associated with these sites. Qualitative interviews will be carried out by the University of Edinburgh.

### Inclusion criteria
► Diagnosis of T1D (WHO criteria) within previous 21 days.
► At least 10 years old and not older than 16.9 years.
► Participant/carer willing to perform regular capillary blood glucose monitoring (at least four blood glucose measurements every day).
► Literate in English.
► Willing to wear study devices.
► Willing to follow study-specific instructions.
► Willing to upload pump and continuous glucose monitoring (CGM) data at regular intervals.

### Exclusion criteria
► Physical/psychological disease likely to interfere with the normal conduct of the study and interpretation of the study results as judged by the investigator.
► Current treatment with drugs known to interfere with glucose metabolism.
► Known or suspected allergy to insulin.
► Regular use of acetaminophen.
► Lack of reliable telephone facility for contact.
► Pregnancy, planned pregnancy or breast feeding.
► Living alone.
► Severe visual or hearing impairment.
► Medically documented allergy towards the adhesive of plasters or unable to tolerate tape adhesive in the area of sensor placement.
► Serious skin diseases located at places of the body used for localisation of the glucose sensor.
► Illicit drugs abuse.
► Prescription drugs abuse.
► Alcohol abuse.
► Sickle cell disease, haemoglobinopathy, receiving red blood cell transfusion or erythropoietin within 3 months prior to time of screening.
► Eating disorder including anorexia/bulimia.
► Milk protein allergy.

### Study schedule
The study will consist of up to 14 visits and one telephone/email contact in each arm over the 24-month study period.

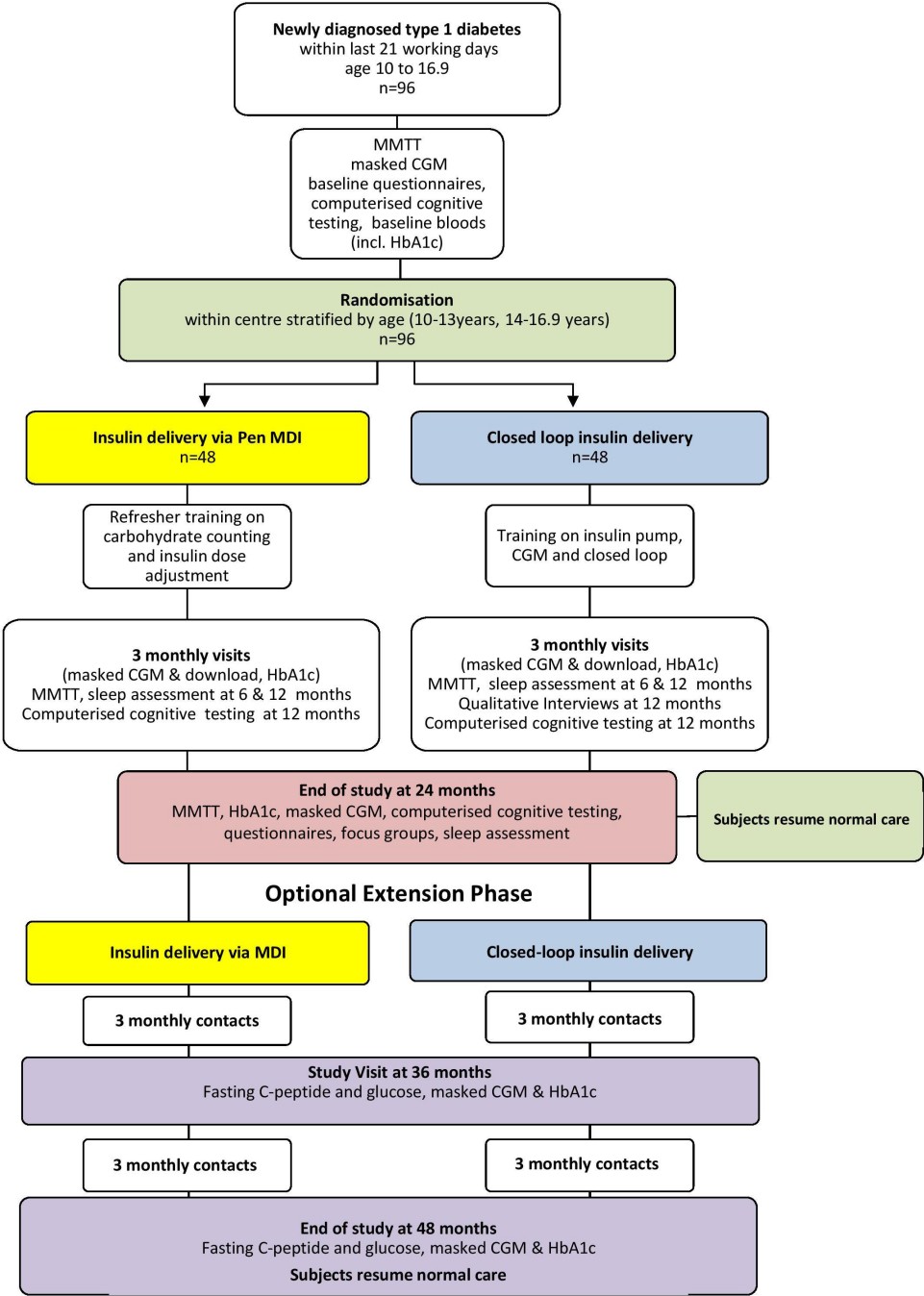

**Figure 1** Study flow including extension phase. CGM, continuous glucose monitoring; HbA1c, glycated haemoglobin; MDI, multiple daily injection; MMTT, mixed meal tolerance test

The 24-month extension phase includes three visits and six clinic/telephone/email contacts (tables 1 and 2).

Following diagnosis and screening, written informed consent or assent (for minors) will be obtained from all participants and guardians before any study-related activities.

### Run-in and prerandomisation training
Participants and their families will receive structured diabetes education and training as per usual clinical practice in accordance with ISPAD guidelines.[22] All participants will be trained on the multiple daily injection (MDI) regimen (online supplementary appendix).

Within 3 weeks of diagnosis, participants will have a baseline mixed meal tolerance test (MMTT).

### Randomisation
Eligible subjects will be randomised after baseline MMTT using remote central randomisation software to the use of the Cambridge CL insulin delivery system or to standard therapy. Randomisation will be stratified by site and age. The randomisation ratio will be 1:1 within each stratum. The randomisation list created by the study statistician is encrypted.

**Table 1**  Schedule of study visits and contacts when the participant is randomised to day-and-night closed-loop (intervention group)

| | Visit/ contact | Description | Start relative to previous/next visit/activity | Duration |
|---|---|---|---|---|
| Run in period | Visit 1 | Recruitment and screening visit: consent/assent; inclusion, exclusion; screening blood sample | Within 21 days of diagnosis | 2 hours |
| | Visit 2 | Baseline visit: HbA1c, MMTT, masked CGM, questionnaires, computerised cognitive testing, bloods for immunological analyses | 7–21 days after diagnosis | 3–4 hours |
| Randomisation | | | | |
| Insulin pump and CGM training | Visit 3 | Insulin pump training, initiation study pump | Within 1 week of Visit 2 | 3–4 hours |
| | Visit 4 | CGM training, initiation of CGM | Within 0–7 days of Visit 3 (Visit 4 may coincide with Visit 3; training visits can be repeated) | 2 hours |
| Closed-loop insulin delivery (24 months) | *Visit 5 | Closed-loop initiation at clinic/home | Within 6 weeks of diagnosis | 3–4 hours |
| | Contact | Review use of study devices, study update | 1 week after Visit 5 (±3 days) | <0.5 hour |
| | *Visit 6 | HbA1c, data download, masked CGM | After 3 months of diagnosis (±1 week) | <1 hour |
| | Visit 7 | MMTT, HbA1c, bloods for immunological analyses, data download, masked CGM, sleep quality assessment | After 6 months of diagnosis (±2 weeks) | 3–4 hours |
| | *Visit 8 | HbA1c, data download, masked CGM | After 9 months of diagnosis (±2 weeks) | <1 hour |
| | Visit 9 | MMTT, HbA1c, bloods for immunological analyses, data download, masked CGM, questionnaires, computerised cognitive testing, interviews, sleep quality assessment | After 12 months of diagnosis (±2 weeks) | 3–4 hours |
| | *Visit 10 | HbA1c, data download, masked CGM | After 15 months of diagnosis (±2 weeks) | <1 hour |
| | *Visit 11 | HbA1c, data download, masked CGM | After 18 months of diagnosis (±2 weeks) | <1 hour |
| | *Visit 12 | HbA1c, data download, masked CGM | After 21 months of diagnosis (±2 weeks) | <1 hour |
| | *Visit 13 | Masked CGM, sleep quality assessment | Between Visit 12 and Visit 14 (Visit 13 may coincide with visit 14) | <0.5 hour |
| | Visit 14 | End of closed-loop treatment: MMTT, HbA1c, data download, bloods for immunological analyses, questionnaires, computerised cognitive testing, focus groups | After 24 months of diagnosis (±2 weeks) | 4–5 hours |
| Optional extension phase (24 months) | Contact | Review use of study devices, HbA1c, study update | 3 months after Visit 14 (±2 weeks) | <0.5 hour |
| | Contact | Review use of study devices, HbA1c, study update | 6 months after Visit 14 (±2 weeks) | <0.5 hour |
| | Contact | Review use of study devices, HbA1c, study update | 9 months after Visit 14 (±2 weeks) | <0.5 hour |
| | Visit 15 | Fasted C-peptide and glucose, HbA1c, masked CGM, questionnaires | After 36 months of diagnosis (±2 weeks) | <1 hour |
| | Contact | Review use of study devices, HbA1c, study update | 3 months after Visit 15 (±2 weeks) | <0.5 hour |
| | Contact | Review use of study devices, HbA1c, study update | 6 months after Visit 15 (±2 weeks) | <0.5 hour |
| | Contact | Review use of study devices, HbA1c, study update | 9 months after Visit 15 (±2 weeks) | <0.5 hour |
| | *Visit 16 | Masked CGM | 2 weeks before Visit 17 (±2 weeks) | <0.5 hour |
| | Visit 17 | Fasted C-peptide and glucose, HbA1c, masked CGM review, questionnaires | After 48 months of diagnosis (±2 weeks) | <1 hour |

*Could be done at home.
CGM, continuous glucose monitoring; HbA1c, glycated haemoglobin; MMTT, mixed meal tolerance test.

**Table 2** Schedule of study visits/phone contacts when the participant is randomised to standard therapy that is, multiply daily injections (control group)

| | Visit/ contact | Description | Start relative to previous/next visit/activity | Duration |
|---|---|---|---|---|
| Run in period | Visit 1 | Recruitment and screening visit: consent/assent; inclusion, exclusion; screening blood sample | Within 21 days of diagnosis | 2 hours |
| | Visit 2 | Baseline visit: HbA1c, MMTT, masked CGM, questionnaires, computerised cognitive testing, bloods for immunological analyses | 7–21 days after diagnosis | 3–4 hours |
| Randomisation | | | | |
| Additional Training | Visit 3 | Training on carbohydrate counting | Within 1 week of Visit 2 | 2 hours |
| | Visit 4 | Training on insulin dose adjustment | Within 0–7 days of Visit 3 (Visit 4 may coincide with Visit 3; Training visits can be repeated) | 2 hours |
| Multiple daily injection of insulin (24 months) | *Visit 5 | MDI arm start visit | Within 6 weeks of diagnosis | <1 hour |
| | Contact | Study update | 1 week after Visit 5 (±3 days) | <0.5 hour |
| | *Visit 6 | HbA1c, masked CGM | After 3 months of diagnosis (±1 week) | <1 hour |
| | Visit 7 | MMTT, HbA1c, bloods for immunological analyses, masked CGM, sleep quality assessment | After 6 months of diagnosis (±2 weeks) | 3–4 hours |
| | *Visit 8 | HbA1c, masked CGM | After 9 months of diagnosis (±2 weeks) | <1 hour |
| | Visit 9 | MMTT, HbA1c, bloods for immunological analyses, masked CGM, questionnaires, computerised cognitive testing, sleep quality assessment | After 12 months of diagnosis (±2 weeks) | 3–4 hours |
| | *Visit 10 | HbA1c, masked CGM | After 15 months of diagnosis (±2 weeks) | <1 hour |
| | *Visit 11 | HbA1c, masked CGM | After 18 months of diagnosis (±2 weeks) | <1 hour |
| | *Visit 12 | HbA1c, masked CGM | After 21 months of diagnosis (±2 weeks) | <1 hour |
| | *Visit 13 | Masked CGM, sleep quality assessment | Between Visit 12 and Visit 14, (may coincide with visit 14) | <1 hour |
| | Visit 14 | End of closed-loop treatment: MMTT, HbA1c, bloods for immunological analyses, questionnaires, computerised cognitive testing, focus groups | After 24 months of diagnosis (±2 weeks) | 4–5 hours |
| Optional extension phase (24 months) | Contact | Study update, HbA1c | 3 months after Visit 14 (±2 weeks) | <0.5 hour |
| | Contact | Study update, HbA1c | 6 months after Visit 14 (±2 weeks) | <0.5 hour |
| | Contact | Study update, HbA1c | 9 months after Visit 14 (±2 weeks) | <0.5 hour |
| | Visit 15 | Fasted C-peptide and glucose, HbA1c, masked CGM, questionnaires | After 36 months of diagnosis (±2 weeks) | <1 hour |
| | Contact | Study update, HbA1c | 3 months after Visit 15 (±2 weeks) | <0.5 hour |
| | Contact | Study update, HbA1c | 6 months after Visit 15 (±2 weeks) | <0.5 hour |
| | Contact | Study update, HbA1c | 9 months after Visit 15 (±2 weeks). | <0.5 hour |
| | *Visit 16 | Masked CGM | 2 weeks before Visit 17 (±2 weeks) | <0.5 hour |
| | Visit 17 | Fasted C-peptide and glucose, HbA1c, masked CGM review, questionnaires | After 48 months of diagnosis (±2 weeks) | <1 hour |

*Could be done at home.
CGM, continuous glucose monitoring; HbA1c, glycated haemoglobin; MDI, multiple daily injection; MMTT, mixed meal tolerance test.

## Postrandomisation training
### Closed-loop
Participants randomised to the CL arm will be trained on the use of the study insulin pump and CGM sensor prior to starting CL insulin delivery. Competency on the use of study devices will be assessed.

Participants will use either the FlorenceM CL system (online supplementary appendix) and/or the follow-up CamAPS FX CL platform (online supplementary appendix). Participants can transit from FlorenceM to CamAPS FX at any time during the study and some

participants will use CamAPS FX from randomisation. The control algorithm is identical in both CL systems.

### Standard therapy (control intervention)

Participants will receive additional training sessions to complement the core training and to match contact time with the CL group. Participants will apply standard insulin therapy during the study period. Participants will be allowed to switch to insulin pump therapy and use flash/continuous glucose monitoring or approved CL systems if clinically indicated applying National Institute for Health and Care Excellence (NICE) criteria according to usual clinical practice.

### Contacts during 24-month home study period

Participants will have identical planned contact visits during the two study arms. Participants/parents will be contacted by email/telephone within 1 week after initiation of the respective study arm. Thereafter, participants will be followed up at 3-monthly intervals. These study visits can take place at the hospital clinic, home or other suitable location. The purpose of these visits is to record any adverse events, device deficiencies, changes in insulin doses, other medical conditions and/or medications. An HbA1c sample will be taken and data from study devices downloaded. At each follow-up visit, participants of both study arms will be fitted with a masked CGM sensor. The sensor will be worn at home for up to 14 days. If the sensor fails or sensor function is interrupted prematurely, another sensor may be applied.

Routine clinical care will be provided by the local paediatric diabetes team as per usual care. Throughout the trial, subjects/parents and/or the clinical team are free to adjust insulin therapy as per usual clinical practice, but no active treatment optimisation will be undertaken by the study team.

### Participant withdrawal criteria

The following prerandomisation withdrawal criteria will apply:

1. Participant/family member is unable to demonstrate safe application of MDI therapy during run-in period as judged by the investigator

The following prerandomisation and postrandomisation withdrawal criteria will apply:

2. Participant is unable to demonstrate safe use of MDI or study insulin pump and/or CGM during postrandomisation training as judged by the investigator.
3. Participant fails to demonstrate compliance with MDI therapy or study insulin pump and/or CGM during postrandomisation training.
4. Participants may terminate participation in the study at any time without necessarily giving a reason and without any personal disadvantage.
5. Significant protocol violation or non-compliance.
6. Recurrent severe hypoglycaemia events not related to the use of the CL system.

7. Recurrent severe hyperglycaemia event/DKA unrelated to infusion site failure and related to the use of the CL system.
8. Decision by the investigator or Sponsor that termination is in the participants best medical interest.
9. Allergic reaction to insulin.
10. Allergic reaction to adhesive surface of infusion set or glucose sensor.
11. If patient cannot be contacted in 12 weeks they will be considered lost to follow-up.

Withdrawn participants due to reasons 5–11 will be invited to undergo MMTT and provide blood samples at the end of the planned study intervention.

## STUDY PROCEDURES

### Mixed meal tolerance test

MMTTs will be conducted at baseline, 6, 12 and 24 months post diagnosis. MMTT will commence following an overnight fast provided the participant's blood glucose level is between 4 and 11.1 mmol/L. Long-acting insulin or basal infusion rates will continue as normal. Participants will be given a liquid meal (Boost, Nestle, Switzerland) according to bodyweight. Venous blood samples for measurement of C-peptide and plasma glucose will be collected 10 min prior to the meal, and at 0, 15, 30, 60, 90 and 120 min.

### Blood samples

Blood sampling schedule is shown in online supplementary appendix. HbA1c will be measured centrally (Swansea University, UK) using an International Federation of Clinical Chemistry and Laboratory Medicine aligned method. Total cholesterol, triglycerides, high-density and low-density lipoproteins (HDL and LDL) will be measured locally. Plasma samples for C-peptide and glucose will be processed locally and stored deep frozen (−20°C or below) until analysis at central laboratory (Swansea University, UK). Blood samples for analysis of immune markers from plasma and peripheral mononuclear cells will be analysed centrally (JDRF/Wellcome Trust Diabetes and Inflammation Laboratory, Oxford, UK).

### Masked CGM

Masked CGM (FreeStyle Libre Pro sensor; Abbott Diabetes Care, Alameda, California, USA) will be intermittently applied throughout the trial. During run-in, masked CGM will serve to gain knowledge of the participant's glucose control characteristics before beginning either intervention arm. Postrandomisation, masked CGM will be applied following each 3-monthly study visit for 24 months. During the extension phase masked CGM will be applied every 12 months.

### Psychosocial assessments

#### Questionnaires

Participants and guardians will be invited to complete a series of questionnaires regarding quality of life at

baseline, 6, 12, 24, 36 and 48 months (details in online supplementary appendix).

### Computerised cognitive testing
Participants will complete selected subtests from a computerised test battery, Cogstate[23] at baseline, 12 and 24 months (online supplementary appendix). Cogstate has demonstrated sensitivity to subtle changes in cognition and is designed to accommodate repeated assessment of a single individual.

### Measures of sleep quality
Quality and duration of sleep will be assessed subjectively in participants (Pittsburgh Sleep Quality Index (PSQI) and sleep diary) and guardians (PSQI). Participants will wear an Actiwatch (Philips Respironics, Bend, Oregon, USA) to provide objective measures of sleep and wakefulness based on motor activity. These measures will be conducted at 6, 12 and 24 months.

### Interviews
An integrated qualitative substudy will explore parents' and youth's views about using CL systems; the impact of using CL systems on diabetes management practices and everyday family life; parents' and youth's information and support needs when using CL systems; and their views about how the technology could be improved for future users. In-depth interviews will be undertaken at 12 months with a subset of up to 20 youth and 20 parents in the CL arm. Purposive sampling will be used to ensure diversity in terms of youth's age and gender, parents' occupation/ education and family forms. Where possible, parents and youth from the same family will be interviewed. Participants will be interviewed separately unless a joint interview is requested. Recruitment to the interview study will continue until data saturation is achieved.

### Health economics
Health economic analysis will be performed contrasting CL and MDI therapy using a health economic simulation model: the IQVIA Core Diabetes Model (CDM). The CDM is a validated non-product-specific policy analysis tool for cost-effectiveness analysis in T1D.[24] Baseline characteristics of the simulation cohort will come from the trial. Treatment effects, both on risk factors and specific quality of life, will be based on the trial findings at 12 months for both arms. The base-case analysis will be performed from the perspective of the UK National Health Service. For treatment costs, only the incremental costs between the two arms (CL vs MDI) will be considered.

## PATIENT AND PUBLIC INVOLVEMENT
The research question and study endpoints are based on feedback from participants of previous studies and in line with prioritisation by stakeholders. Study design and assessment of the burden of the intervention were reviewed by focus groups. Results will be disseminated to participants and general public through social media and will be made available on the sponsor's website.

## STATISTICAL ANALYSIS
All analyses will be conducted on an intention-to-treat basis. Data from all randomised participants with/ without protocol violation including dropouts and withdrawals will be included in the analysis. The statistical analysis plan can be found in the online supplementary information.

### Primary outcome analysis
The primary analysis will evaluate between group differences in the mean stimulated C-peptide AUC at 12 months post diagnosis. The values will be compared using a linear model adjusting for baseline log (C-peptide AUC +1), gender, presence/absence of DKA at diagnosis and age as fixed effects, and clinical site as a random effect. HbA1c levels, body mass index (BMI) z-score and race/ ethnicity may also be included in the model as fixed effects to assess for the presence of confounding. The mean-adjusted difference between treatment groups and the corresponding 95% CI from the linear model will be reported. If residual values from the regression model have a skewed distribution then an appropriate alternate transformation or a non-parametric analysis based on ranks will be performed. Primary analysis will be a single comparison and no attempt will be formally made to control the overall type I error rate. The primary outcome will be tested at $\alpha=0.04$.

### Key secondary endpoints
The following outcomes will be compared between treatment groups at 12 months post diagnosis:
► Percent time in the target range (3.9–10.0 mmol/L).
► HbA1c.
► Percent time below 3.9 mmol/L.

These three endpoints will be tested in a hierarchical fashion in the order listed above. If C-peptide AUC is significant at $\alpha=0.04$, then they will be tested at $\alpha=0.05$. Otherwise, they will be tested at $\alpha=0.01$. Likewise, if all three key endpoints are significant at $\alpha=0.01$, then this alpha is recycled to the primary outcome and C-Peptide AUC will be tested at $\alpha=0.05$. This process controls for the family-wise error rate. A diagram displaying this process is shown in the online supplementary appendix.

### Secondary efficacy endpoints
The following outcomes will be compared between treatment groups at 6, 12 and 24 months post diagnosis and at 36 and 48 months for subjects enrolling in the extension phase of the study.
► Mean stimulated C-peptide AUC (24 months).
► Fasting C-peptide/fasting glucose (36 and 48 months).
► HbA1c outcomes:
  – HbA1c levels.

- – Percentage of patients in each group with HbA1c<7.5% (58 mmol/mol).
- ► CGM outcomes*
  - – Percentage of time with sensor glucose readings in target (3.9 to 10 mmol/L).
  - – Mean, SD, and coefficient of variation of sensor glucose levels.
  - – Percentage of time with sensor glucose above target (10.0 mmol/L).
  - – Percentage of time with sensor glucose levels in significant hyperglycaemia (>16.7 mmol/L).
  - – Percentage of time spent below target glucose (3.9 mmol/L).
  - – Percentage of time with sensor glucose levels<3.5 mmol/L,<3.0 mmol/L and <2.8 mmol/L.
  - – AUC of sensor glucose below 3.9 mmol/L and 3.5 mmol/L.
- ► Total, basal and bolus insulin dose (units/day/kg).
- ► Change in BMI SD score.
- ► Blood pressure.
- ► Lipid profile.

*Glycaemic metrics will be based on masked sensor glucose levels collected at 3-monthly intervals during the trial. A single percentage will be calculated for each subject at each visit by pooling all CGM readings corresponding to the 14-day period.

For all secondary endpoints, the false discovery rate will be controlled using the adaptive Benjamini-Hochberg procedure.

### Safety analysis
The following events will be compared between treatment groups:
- ► Number of subjects with severe hypoglycaemic events.
- ► Number of episodes of severe hypoglycaemic events per subject and incidence rate/100-person years
- ► Number of subjects with Diabetic Ketoacidosis (DKA) events.
- ► Number of episodes of DKA events per subject and incidence rate/100-person years.
- ► Number of any other adverse events reported per subject.
- ► Number of any other severe adverse events reported per subject.

Safety data will be tabulated for all subjects in the two intervention periods, including dropouts and withdrawals, irrespective of whether sensor glucose data are available and irrespective of whether CL was operational. For the binary safety outcomes, a repeated measures logistic regression model will be used to compare treatment arms. For the count outcomes and the incidence rates, a Poisson regression model will be used.

### Psychosocial evaluation
#### Questionnaires
For each questionnaire, mean±SD score for each dimension and total score will be tabulated by treatment group, in addition to the distribution of responses for each individual question for both participant and parent version. The between-group difference of each score will be assessed using a linear model, adjusting for age, gender, presence of DKA at diagnosis and corresponding score at baseline, and adjusting for clinical site as a random effect.

### Cognitive testing
Change in cognitive test performance from baseline to follow-up at 12 and 24 months will be assessed, as a function of group (intervention or control).

### Sleep quality
The PSQI and actigraph data will be used to calculate mean total sleep quality score, sleep duration (sum of all epochs scored as sleep during the time in bed) and variability across nights, time in bed, sleep disturbance (including wake after sleep onset and number of awakenings), latency, efficiency, quality and daytime dysfunction. Sleep data will be averaged across nights in each participant for each study period.

### Qualitative interview
Interview data will be analysed thematically using the constant comparison method. NVivo V.9, a qualitative software package, will be used to facilitate data coding/retrieval.

### Focus groups
Transcripts of focus group discussions will be thematically analysed using QSR NVivo qualitative analysis software.

Protocol adherence will be analysed by treatment group.

### Health economics
Long-term outcomes derived from the simulation will include total direct costs, life expectancy, quality-adjusted life expectancy and time to onset of complications. Incremental costs versus incremental effectiveness (quality-adjusted life years) for CL versus MDI therapy will be compared.

### Subgroup analysis
A random centre by treatment interaction effect will be explored for the primary outcome. Interpretation of any subgroup analyses will depend on whether the overall analysis demonstrates a significant treatment group difference. In the absence of a significant treatment effect in the primary analysis, interpretation of the centre by treatment interaction will be considered exploratory.

### Per-protocol analysis
Per-protocol analysis limited to participants in the CL group who used CL for ≥60% of the time and in the MDI group who do not start insulin pump therapy will be conducted to compare the primary outcome, HbA1c and time with sensor glucose levels in target range.

### Interim analysis
No formal interim analyses are planned.

## Power calculation

Assuming a mean area under the meal-stimulated C-peptide curve of 0.37 pmol/mL for the control group based on the lower 90% confidence limit from previous data,[25] a 50% increase in the intervention group gives 0.37*1.50=0.555 pmol/mL. After a ln(x+1) transformation, the mean values in the control and treatment groups are 0.315 and 0.441, respectively, giving a treatment effect of 0.126. The treatment effect of 0.126 with an SD of 0.18 requires 44 subjects per group at 90% power for a two-sided test at the 0.05 level. Allowing for 10% loss to follow-up means we would need a total of 96 randomised participants.

## STUDY MANAGEMENT

Composition of study management groups is shown in the online supplementary appendix.

### Data Monitoring and Ethics Committee

An independent Data Monitoring and Ethics Committee (DMEC) will be notified of all serious adverse events and any unanticipated adverse device events that occur during the study. The DMEC will review compiled adverse event data at periodic intervals and will report to the Trial Steering Committee any safety concerns and recommendations for suspension or early termination of the trial.

### Study sponsor

The study sponsors are the Cambridge University Hospitals NHS Foundation Trust and the University of Cambridge.

### Trial Steering Committee

The trial steering committee will meet biannually to provide overall supervision of the trial including progress of the trial, adherence to the protocol, patient safety and the consideration of new information of relevance to the research question.

### Trial Management Group

The trial management group (TMG) will meet weekly and will be responsible for day-to-day management of the trial.

### Data management and monitoring

The study coordinators will be responsible for maintaining quality assurance and quality control systems to ensure that the trial is conducted and data are generated, documented and reported in compliance with the protocol, good clinical practice and regulatory requirements.

Confidentiality of participant data shall be observed at all times. Personal details for each participant taking part with a link to a unique identification number will be held locally on a study screening log in the Trial Site File at each study site. These details will not be revealed at any other stage during the study, and all results will remain anonymous.

Electronic case report forms will be used for recording anonymised study data and will be completed in accordance with Good Clinical Practice and ISO 15197: 2013 guidelines.

### Indemnity

Any liability arising from study design will be covered by the clinical trial insurance policy organised by the University of Cambridge. National Health Service indemnity cover will apply for any claims arising from management and conduct of research.

## ETHICS AND DISSEMINATION

All participants will be provided with oral and written information about the trial and procedures involved in the study before obtaining written informed consent/assent.

Standard operating procedures for monitoring and reporting of all adverse events and adverse device effects will be in place including serious adverse events, serious adverse device effects and specific adverse events such as severe hypoglycaemia and significant hyperglycaemia with ketosis. Any substantial amendments to the protocol and other documents shall be notified to and approved by the independent REC and the regulatory authorities, prior to implementation as per nationally agreed guidelines.

Screening and recruitment commenced in January 2017, and the study is expected to be completed by October 2023. Study results will be disseminated by peer-reviewed publications and conference presentations.

**Author affiliations**
[1]Wellcome Trust-MRC Institute of Metabolic Science, University of Cambridge, Cambridge, UK
[2]Department of Paediatrics, University of Cambridge, Cambridge, UK
[3]Children's Diabetes Centre, Leeds Children's Hospital, Leeds, UK
[4]Department of Diabetes, Alder Hey Children's NHS Foundation Trust, Liverpool, UK
[5]Department of Paediatric Diabetes and Endocrinology, Nottingham Children's Hospital, Nottingham, UK
[6]NIHR Oxford Biomedical Research Centre, Oxford University Hospitals NHS Foundation Trust, Oxford, UK
[7]Department of Paediatrics, University of Oxford, Oxford, UK
[8]Paediatric Diabetes, Southampton Children's Hospital, Southampton, UK
[9]Department of Diabetes, Royal Hospital for Sick Children, Edinburgh, UK
[10]Murdoch Children's Research Institute, Parkville, Victoria, Australia
[11]Endocrinology, Stanford University School of Medicine, Stanford, California, USA
[12]Leeds Institute of Cardiovascular and Metabolic Medicine, University of Leeds, Leeds, UK
[13]The University of Edinburgh Usher Institute of Population Health Sciences and Informatics, Edinburgh, UK
[14]HEVA HEOR Sarl, Lyon, France
[15]Jaeb Center for Health Research, Tampa, Florida, USA
[16]Wellcome Trust Centre for Human Genetics, Oxford, UK

**Acknowledgements** Jasdip Mangat supported development and validation of closed-loop system. Cambridge Clinical Trials Unit, Nicole Ashcroft, Josephine Hayes and Matthew Haydock (Institute of Metabolic Science, University of Cambridge) provide administrative support. NIHR Cambridge Clinical Research Facility will support the research team in their research-related activities. Professor Stephen Luzio at the University of Swansea Diabetes Research Unit Cymru will analyse the blood samples. Artificial pancreas focus group contributors provided feedback on the study design.

**Contributors** RH, MEW, CLA, FC, AG, TR, RB, NT, DE and KH codesigned the study. CK and NC designed the statistical plan. CB, JMA, MT, SH, GM, CLA, DE, FC, AG, TR, RB, NT and DE screened and enrolled participants, arranged informed consent from the participants, provided patient care and took samples. KH, EN, JL and ES will conduct the psychosocial assessments. SR conducted the cost utility analysis. JT conducted the immunology assessments. JS coordinated the study. RH designed and implemented the glucose controller. CB and RH wrote the manuscript. All authors critically reviewed the report. No writing assistance was provided.

**Funding** National Institute for Health Research EME Grant (14/23/09) and Leona M & Harry B Helmsley Charitable Trust Grant (#2016PG-T1D046). Additional support for the artificial pancreas work is from National Institute for Health Research Cambridge Biomedical Research Centre, and Wellcome Strategic Award (100574/Z/12/Z). Medtronic and Dexcom are supplying discounted CGM devices, sensors and details of communication protocol to facilitate real-time connectivity. Abbott Diabetes Care is providing Libre Pro sensors. JAT is supported by a strategic award to the Diabetes and Inflammation Laboratory from the JDRF (4-SRA-2017–473-A-A) and the Wellcome Trust (107212/A/15/Z), and a grant from the JDRF (1-SRA-2019–657-A-N).

**Competing interests** RH reports having received speaker honoraria from Eli Lilly and Novo Nordisk, serving on advisory panel for Eli Lilly and Novo Nordisk, receiving licence fees from BBraun and Medtronic. RH and MEW report patient patents and patent applications. MT has received speaker honoraria from Medtronic and Novo Nordisk. SH is a member of Sigma (Dexcom) advisory board and reports having received training honoraria from Medtronic and Sanofi. TLR has received speaker honoraria from Novo Nordisk and serves as a consultant for Abbott Diabetes Care. KH has received research support from Dexcom, Inc for an investigator-initiated project; he has received consultant fees from Lilly Innovation Center, Bigfoot Biomedical, and Insulet, Inc.

**Patient consent for publication** Not required.

**Ethics approval** The study has received approval from the Cambridge East Research Ethics Committee in the UK (#16/EE/0286) and has undergone review by regulatory authorities in the UK (Medicines and Healthcare products Regulatory Agency).

**Provenance and peer review** Not commissioned; externally peer reviewed.

**ORCID iDs**
Charlotte Boughton http://orcid.org/0000-0003-3272-9544
Professor David Dunger http://orcid.org/0000-0002-2566-9304

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
