## [Reviewer comments · BMJ Open]

ARTICLE DETAILS

TITLE (PROVISIONAL)	Assessing the effect of closed loop insulin delivery from onset of type 1 diabetes in youth on residual beta-cell function compared to standard insulin therapy (CLOuD Study): a randomised parallel study protocol
AUTHORS	Boughton, Charlotte; Allen, Janet; Tauschmann, Martin; Hartnell, Sara; Wilinska, Malgorzata; Musolino, Gianluca; Acerini, Carlo; Dunger, Professor David; Campbell, Fiona; Ghatak, Atrayee; Randell, Tabitha; Besser, Rachel; Trevelyan, Nicola; Elleri, Daniela; Northam, Elizabeth; Hood, Korey; Scott, Eleanor; Lawton, Julia; Roze, Stephane; Sibayan, Judy; Kollman, Craig; Cohen, Nate; Todd, John; Hovorka, Roman

VERSION 1 – REVIEW

REVIEWER	Israel Hodish University of Michigan; US I am a co-founder of Hygieia INC.
REVIEW RETURNED	30-Aug-2019

GENERAL COMMENTS	No specific comments.
-----------------------

REVIEWER	Agnieszka Szypowska Medical University of Warsaw, Poland
REVIEW RETURNED	07-Sep-2019

GENERAL COMMENTS	Authors would like to evaluate if the closed-loop system may preserve residual beta-cell function in type 1 adolescents with newly recognized diabetes. To test the hypothesis an open-label, multicenter, randomized clinical trial is performed. The study aims to randomize 96 participants. Recruited participants are randomly assigned to 24 months of study intervention (closed-loop system vs. MDI therapy). The primary outcome is the between group difference in area under the stimulated C-peptide curve (AUC) of the MMTT at 12 months post-diagnosis. The topic is clinically relevant. The research methodology is good. Comments: Line 23. "Percentage of patients in each group with HbA1c <7.5% (58 mmol/mol)..." According to ISPAD guidelines "For children, adolescents, and young adults ≤25 years who have access to comprehensive care a target of HbA1c of <53mmol/mol (7%) is recommended." In your analysis HbA1c target should be <7% instead of <7.5%. 2. In the Pediatric Onset Study comparing CSII+RT-CGM vs. CSII at diabetes onset. Patients with frequent sensor use had significantly less C-peptide reduction within 24 months. The results of the study
---

	indicate an important role for CGM in lowering blood glucose fluctuations and preserving the C-peptide. Currently, many people are treated with MDI but use RT-CGM or FGM. Thanks to this they achieve better glycemic control with lower glucose fluctuations. Looking at the results of earlier studies, it seems obvious that the closed loop vs. MDI + SMBG will result in a longer time in range and a higher C-peptide. In my opinion more important is the answer if closed loop vs. MDI + RT-CGM will affect the C-peptide. I believe that for reliable closed-loop assessment, the control group should use RT-CGM to measure glucose (DexCom 6). Moreover, the protocol should detail how glucose is measured in the MDI group (only SMBG, occasionally RT-CGM, FGM...). 3. Children in MDI group should measure blood glucose (SMBG) more than 4 times/day. It is recommended in ISPAD guidelines: "When fingerstick BGs are used, testing may need to be performed 6 to 10 times per day to optimize intensive control." The number of glucose measurements recommended in the protocol should be consistent with ISPAD recommendations.
--	--

REVIEWER	Benyamin Grosman Medtronic, USA
	The reviewer is an employee and share holder in Medtronic.
REVIEW RETURNED	31-Oct-2019

GENERAL COMMENTS	The protocol submitted is well designed and well written. Maintaining the functionality of beta cell at onset diabetes is important to research. My main remark is about using closed-loop (CL) as a thing. It should be emphasized throughout this protocol that it is not CL in general together with all other available CL algorithm and systems that are available in market or in research platforms, but it is, this specific CL algorithm effectiveness at onset type 1 diabetes that is being evaluated. Moreover, writing an introduction about CL without mentioning the only commercially available system, Medtronic 670G, seems odd. I would expect at least a sentence and a reference to the only commercial available system.
---

REVIEWER	Garry M Steil Boston Children's Hospital
REVIEW RETURNED	06-Nov-2019

GENERAL COMMENTS	General comments The investigators propose to study 96 individuals with new onset T1D randomized to standard care (MDI) or closed-loop insulin delivery (MPC): 48 per group. Primary outcome is AUC of C-peptide response to a mixed meal administered at 12 months. Secondary objectives include reassessment of AUC at 24 months, with an optional 2-year study extension assessing fasting values at 36 and 48 months. Patients will be recruited from 7 sites. Multiple secondary endpoints are defined, all of which are appropriate, and predefined exploratory analysis, also appropriate. The study is registered at ClinicalTrials.gov (NCT02871089) Comments major 1. The paper and/or appendices indicates the first subject in the
--

	study is planned to take place in August 2016 and expected completion of the last subject June 2021. ClinicalTrials.gov lists the study as “Active, not recruiting” with start date Jan 2017 and estimated completion Oct 2021. This is inconsistent. Studies are registered so that the final methods and results can be compared to what was planned at the study start, with inconsistencies obviously raising red flags. The manuscript should reference the date at which ClinicalTrials.gov site information was last updated and make sure the information in the manuscript/appendices is consistent with what was reported at the time the manuscript was written. 2. The protocol/trial registration indicates that up to 190 subjects are “expected” to be recruited with 96 subjects expected to be randomized. Figure 2 indicates that between recruitment and randomization the investigators will perform a mixed meal tolerance test (MMTT), blinded CGM, baseline questionnaires, computerised cognitive testing, and baseline bloods (incl. HbA1c). It is unclear how the investigators plan to use the information obtained in these tests and/or questionnaires to exclude ½ of the patients from randomization. Moreover, having the ability to do so leaves the investigators substantial latitude to select specific patients not representative of the overall population. 3. There are several problems/concerns related to the power calculation. Specifically: a. Primary outcome is 2-hour AUC of the C-peptide response to MMTT. The authors report the lower 90% confidence limit for AUC obtained in a previous study as 0.37 pmol/ml for control group. However, the units for AUC should be “concentration” times “time”; the reported value is concentration only. b. The study is powered to detect a 50% increase in the intervention group. The authors report that the data are to be log transformed and calculate a treatment effect of 0.126. They then report that for this effect size, a standard deviation of 0.18 would result in 44 subjects per group to obtain 90% power for a two-sided test at the 0.05 level. It’s unclear where 0.18 standard deviation was obtained. Also, the manuscript reports the primary outcome will be tested at $p=0.04$. c. Based on effect size and SD they report the number of subjects per group as 44. This is the correct number for an un-paired t-test of the final mean AUC values obtained. However, the test does not adjust for any potential difference in AUC at baseline; i.e., baseline values are not used in any way. The authors should consider either testing the change in AUC observed in each subject (final – baseline) or move to a 2-way ANOVA with time (repeated) and AUC (between subject) tested. The authors can stay with analysis as described; however, they cannot assess significance using that analysis and, in the event that the treatment effect is not significant, re-test the data using alternate approach without adjusting for multiple comparisons (2 in this case). d. The investigators allowing for 10% loss to follow up. This seems optimistic given that there are numerous reports of high drop-out rates in patients using the Medtronic 670G closed-loop system (in paediatric subjects more than ½ discontinue use before 2 years). e. Two different closed-loop control systems using the same
--	--

	algorithm, with the patient allowed to switch between systems, are to be used in the study. It is not clear if the statistical analysis for the primary outcome will, or should, include sensor and/or control hardware as a variable. If so, it should be stated before the investigators look at the data. 4. Subjects using multiple daily injections (MDI) therapy are used as control subjects. Thus, the study will not be able to determine if closed-loop therapy would lead to a better outcome if the comparative group were to use a more advanced technology such as the Bluetooth Enabled Insulin Pen. These pens allow meal insulin boluses and/or correction boluses to be calculated using an APP. Choosing the least advanced open-loop therapy for the control arm predisposes the study to show a positive effect in the closed-loop arm. The investigators should consider provide study subjects randomized to the control are the option of using a better – or best - open-loop technology. Minor 1. It is unclear why “Regular use of acetaminophen” is an exclusion criterion. Historically, glucose sensors reacted to acetaminophen thereby giving false readings; however, this problem has been solved.
--	--

VERSION 1 – AUTHOR RESPONSE

Reviewer: 1

Please leave your comments for the authors below

No specific comments

We thank the Reviewer for their time in reviewing the manuscript

Reviewer: 2

Please leave your comments for the authors below

Authors would like to evaluate if the closed-loop system may preserve residual beta-cell function in type 1 adolescents with newly recognized diabetes. To test the hypothesis an open-label, multicenter, randomized clinical trial is performed. The study aims to randomize 96 participants. Recruited participants are randomly assigned to 24 months of study intervention (closed-loop system vs. MDI therapy). The primary outcome is the between group difference in area under the stimulated C-peptide curve (AUC) of the MMTT at 12 months post-diagnosis. The topic is clinically relevant. The research methodology is good.

We thank the Reviewer for these positive comments of our work

Comments:

Line 23. “Percentage of patients in each group with HbA1c <7.5% (58 mmol/mol)...” According to ISPAD guidelines “For children, adolescents, and young adults ≤25 years who have access to comprehensive care a target of HbA1c of <53mmol/mol (7%) is recommended.” In your analysis HbA1c target should be <7% instead of <7.5%.

We thank the Reviewer for this suggestion and will include Percentage of patients in each group with HbA1c <7.0% (53 mmol/mol) to the statistical analysis plan in line with the most recent ISPAD guidelines.

2. In the Pediatric Onset Study comparing CSII+RT-CGM vs. CSII at diabetes onset. Patients with frequent sensor use had significantly less C-peptide reduction within 24 months. The results of the study indicate an important role for CGM in lowering blood glucose fluctuations and preserving the C-peptide. Currently, many people are treated with MDI but use RT-CGM or FGM. Thanks to this they achieve better glycemic control with lower glucose fluctuations.

Looking at the results of earlier studies, it seems obvious that the closed loop vs. MDI + SMBG will result in a longer time in range and a higher C-peptide. In my opinion more important is the answer if closed loop vs. MDI + RT-CGM will affect the C-peptide. I believe that for reliable closed-loop assessment, the control group should use RT-CGM to measure glucose (DexCom 6). Moreover, the protocol should detail how glucose is measured in the MDI group (only SMBG, occasionally RT-CGM, FGM...).

The Reviewer makes an important point. The study aims to compare closed-loop glucose control with standard clinical practice, which in the UK at the time of diagnosis is MDI and finger-stick glucose monitoring. The study design is pragmatic due to the duration of the study and the rapid increase in diabetes technology availability and uptake. As such the protocol allows for individualised treatment plans on a person by person basis in accordance with UK clinical guidelines; participants in the control (MDI) group are permitted to use continuous glucose monitoring or Flash glucose monitoring as per standard clinical guidelines (and including those wishing to self-fund). Participants in the control (MDI) arm are also allowed to use pump therapy and even use approved hybrid closed-loop systems in accordance with standard clinical guidelines.

A recently published UK study did not demonstrate superiority of pump therapy alone over MDI at diagnosis in children and adolescents (Blair et al BMJ 2019). Current UK guidelines do not recommend continuous or flash glucose monitoring from diagnosis.

3. Children in MDI group should measure blood glucose (SMBG) more than 4 times/day. It is recommended in ISPAD guidelines: "When fingerstick BGs are used, testing may need to be performed 6 to 10 times per day to optimize intensive control." The number of glucose measurements recommended in the protocol should be consistent with ISPAD recommendations.

We agree with the Reviewer that optimal glycaemic control may require finger-stick BG monitoring 6 to 10 times per day and the protocol states participants must be willing to perform a minimum of 4 measurements per day. There is no upper limit of finger-stick BG measurements. In practice, this intensity of monitoring is not always adhered to, particularly in adolescents. The pragmatic study design supports generalisability of the findings.

Reviewer: 3

Please leave your comments for the authors below

The protocol submitted is well designed and well written. Maintaining the functionality of beta cell at onset diabetes is important to research.

My main remark is about using closed-loop (CL) as a thing. It should be emphasized throughout this protocol that it is not CL in general together with all other available CL algorithm and systems that are available in market or in research platforms, but it is, this specific CL algorithm effectiveness at onset type 1 diabetes that is being evaluated.

Moreover, writing an introduction about CL without mentioning the only commercially available system, Medtronic 670G, seems odd. I would expect at least a sentence and a reference to the only commercial available system.

We thank the Reviewer for these positive comments of the study protocol. At the time of study conception and protocol design there were no commercially available closed-loop systems. For clarity we have added throughout the manuscript that we use the Cambridge closed-loop system.

We have added a sentence to the introduction to mention the current commercially available closed-loop system (Page 7, Paragraph 1).

Reviewer: 4

Please leave your comments for the authors below

General comments

The investigators propose to study 96 individuals with new onset T1D randomized to standard care (MDI) or closed-loop insulin delivery (MPC): 48 per group. Primary outcome is AUC of C-peptide response to a mixed meal administered at 12 months. Secondary objectives include reassessment of AUC at 24 months, with an optional 2-year study extension assessing fasting values at 36 and 48 months. Patients will be recruited from 7 sites. Multiple secondary endpoints are defined, all of which are appropriate, and predefined exploratory analysis, also appropriate. The study is registered at ClinicalTrials.gov (NCT02871089)

We thank the Reviewer for these positive comments of our work

Comments major

1. The paper and/or appendices indicates the first subject in the study is planned to take place in August 2016 and expected completion of the last subject June 2021. ClinicalTrials.gov lists the study as “Active, not recruiting” with start date Jan 2017 and estimated completion Oct 2021. This is inconsistent. Studies are registered so that the final methods and results can be compared to what was planned at the study start, with inconsistencies obviously raising red flags. The manuscript should reference the date at which ClinicalTrials.gov site information was last updated and make sure the information in the manuscript/appendices is consistent with what was reported at the time the manuscript was written.

We thank the Reviewer for this observation. The first subject in the study was recruited in January 2017 and estimated completion is October 2023. We have amended the manuscript to be consistent with the protocol. We have also added last updated information to the manuscript abstract.

2. The protocol/trial registration indicates that up to 190 subjects are “expected” to be recruited with 96 subjects expected to be randomized. Figure 2 indicates that between recruitment and randomization the investigators will perform a mixed meal tolerance test (MMTT), blinded CGM, baseline questionnaires, computerised cognitive testing, and baseline bloods (incl. HbA1c). It is unclear how the investigators plan to use the information obtained in these tests and/or questionnaires to exclude ½ of the patients from randomization. Moreover, having the ability to do so leaves the investigators substantial latitude to select specific patients not representative of the overall population.

We thank the Reviewer for this, and we apologise for this oversight. The 190 refers to those expected to be approached for recruitment into the study. It was anticipated that recruitment to a research study so soon after diagnosis may be challenging, especially with the mixed meal tolerance test and therefore a conservative 50% uptake of those approached was expected.

3. There are several problems/concerns related to the power calculation. Specifically:

a. Primary outcome is 2-hour AUC of the C-peptide response to MMTT. The authors report the lower 90% confidence limit for AUC obtained in a previous study as 0.37 pmol/ml for control group. However, the units for AUC should be “concentration” times “time”; the reported value is concentration only.

The reviewer is correct that the metric being assessed in this study is not truly an AUC. Instead, it is

normalized by the duration of the MMTT (2 hours) so that it reflects a weighted average of the C-peptide measurements. This is commonly done in these studies (Lachin et al. 2011):

Lachin JM, McGee PL, Greenbaum CJ, Palmer J, Pescovitz MD, Gottlieb P, Skyler JS, Type 1 Diabetes Trial Network. 2011. Type 1 Diabetes Trial Network Sample size requirements for studies of treatment effects on beta-cell function in newly diagnosed type 1 diabetes. *PLoS ONE* 6: e26471.

b. The study is powered to detect a 50% increase in the intervention group. The authors report that the data are to be log transformed and calculate a treatment effect of 0.126. They then report that for this effect size, a standard deviation of 0.18 would result in 44 subjects per group to obtain 90% power for a two-sided test at the 0.05 level. It's unclear where 0.18 standard deviation was obtained. Also, the manuscript reports the primary outcome will be tested at $p=0.04$.

The paper mentioned above by Lachin et al. (2011) reports the standard deviation of the $\log(\text{AUC}+1)$ to be 0.167; we used 0.18 as a conservative estimate.

The initial sample size calculation was done at $\alpha=0.05$. Afterwards, it was decided to amend the SAP to adjust for multiple comparisons for C-peptide AUC, HbA1c, time in range, and time <3.9 mmol/L using a hierarchical procedure. Specifically, the primary outcome will be tested at $\alpha=0.04$. If the primary outcome is significant at $\alpha=0.04$, then the other three key endpoints will be tested in a hierarchy at $\alpha=0.05$; otherwise, they will be tested at $\alpha=0.01$. This hierarchical procedure has minimal impact on power for the primary outcome reducing it from 90% in the initial sample size calculation to 88% with the multiple comparisons adjustment.

c. Based on effect size and SD they report the number of subjects per group as 44. This is the correct number for an un-paired t-test of the final mean AUC values obtained. However, the test does not adjust for any potential difference in AUC at baseline; i.e., baseline values are not used in any way. The authors should consider either testing the change in AUC observed in each subject (final – baseline) or move to a 2-way ANOVA with time (repeated) and AUC (between subject) tested. The authors can stay with analysis as described; however, they cannot assess significance using that analysis and, in the event that the treatment effect is not significant, re-test the data using alternate approach without adjusting for multiple comparisons (2 in this case).

The primary analysis will adjust for baseline AUC, but we ignored this in the sample size calculation because we were not confident about how strong the correlation would be one year later. To be conservative, we assumed this correlation would be zero (i.e, ignored baseline in the sample size calculation).

d. The investigators allowing for 10% loss to follow up. This seems optimistic given that there are numerous reports of high drop-out rates in patients using the Medtronic 670G closed-loop system (in paediatric subjects more than ½ discontinue use before 2 years).

The Medtronic 670G closed-loop system was not commercially available at the time of study design and power calculation. Different closed-loop systems have different usability. As the analysis is by intention to treat, even those not using the closed-loop system but remaining in the study will be included in the analysis.

e. Two different closed-loop control systems using the same algorithm, with the patient allowed to switch between systems, are to be used in the study. It is not clear if the statistical analysis for the primary outcome will, or should, include sensor and/or control hardware as a variable. If so, it should be stated before the investigators look at the data.

We will not do any formal statistical comparison of these two systems because there could potentially be a selection bias in which system each person chooses. Therefore, the primary analysis will be a comparison of randomization groups.

4. Subjects using multiple daily injections (MDI) therapy are used as control subjects. Thus, the study will not be able to determine if closed-loop therapy would lead to a better outcome if the comparative group were to use a more advanced technology such as the Bluetooth Enabled Insulin Pen. These pens allow meal insulin boluses and/or correction boluses to be calculated using an APP. Choosing the least advanced open-loop therapy for the control arm predisposes the study to show a positive effect in the closed-loop arm. The investigators should consider provide study subjects randomized to the control are the option of using a better – or best -open-loop technology.

The Reviewer makes an important point. The study aims to compare closed-loop glucose control with standard clinical practice, which in the UK at the time of diagnosis is MDI and finger-stick glucose monitoring. The study design is pragmatic due to the duration of the study and the rapid increase in diabetes technology availability and uptake. As such the protocol allows for individualised treatment plans on a person by person basis in accordance with UK clinical guidelines; participants in the control (MDI) group are permitted to use continuous glucose monitoring or Flash glucose monitoring as per standard clinical guidelines (and including those wishing to self-fund). Participants in the control (MDI) arm are also allowed to use Bluetooth Enabled Insulin Pens, pump therapy and even use approved hybrid closed-loop systems in accordance with standard clinical guidelines. Current UK guidelines do not recommend Bluetooth Enabled Insulin Pens from diagnosis.

Minor

1. It is unclear why “Regular use of acetaminophen” is an exclusion criterion. Historically, glucose sensors reacted to acetaminophen thereby giving false readings; however, this problem has been solved.

The early closed-loop platform used in this study used continuous glucose monitors which were potentially affected by acetaminophen.

VERSION 2 – REVIEW

REVIEWER	Garry Steil Boston USA
REVIEW RETURNED	09-Dec-2019
GENERAL COMMENTS	No further comments.